# Abscisic Acid Mediates Salicylic Acid Induced Chilling Tolerance of Grafted Cucumber by Activating H_2_O_2_ Biosynthesis and Accumulation

**DOI:** 10.3390/ijms232416057

**Published:** 2022-12-16

**Authors:** Yanyan Zhang, Xin Fu, Yiqing Feng, Xiaowei Zhang, Huangai Bi, Xizhen Ai

**Affiliations:** 1State Key Laboratory of Crop Biology, Key Laboratory of Crop Biology and Genetic Improvement of Horticultural Crops in Huanghuai Region, Collaborative Innovation Center of Fruit & Vegetable Quality and Efficient Production in Shandong, College of Horticulture Science and Engineering, Shandong Agricultural University, Tai’an 271018, China; 2Tai’an Academy of Agricultural Sciences, Tai’an 271000, China

**Keywords:** salicylic acid, abscisic acid, hydrogen peroxide, signal transduction chilling tolerance, cucumber, pumpkin

## Abstract

Grafting is widely applied to enhance the tolerance of some vegetables to biotic and abiotic stress. Salicylic acid (SA) is known to be involved in grafting-induced chilling tolerance in cucumber. Here, we revealed that grafting with pumpkin (*Cucurbita moschata*, *Cm*) as a rootstock improved chilling tolerance and increased the accumulation of SA, abscisic acid (ABA) and hydrogen peroxide (H_2_O_2_) in grafted cucumber (*Cucumis sativus/Cucurbita moschata, Cs/Cm*) leaves. Exogenous SA improved the chilling tolerance and increased the accumulation of ABA and H_2_O_2_ and the mRNA abundances of *CBF1*, *COR47*, *NCED*, and *RBOH1*. However, 2-aminoindan-2-phosphonic acid (AIP) and L-a-aminooxy-b-phenylpropionic acid (AOPP) (biosynthesis inhibitors of SA) reduced grafting-induced chilling tolerance, as well as the synthesis of ABA and H_2_O_2_, in cucumber leaves. ABA significantly increased endogenous H_2_O_2_ production and the resistance to chilling stress, as proven by the lower electrolyte leakage (EL) and chilling injury index (CI). However, application of the ABA biosynthesis inhibitors sodium tungstate (Na_2_WO_4_) and fluridone (Flu) abolished grafting or SA-induced H_2_O_2_ accumulation and chilling tolerance. SA-induced plant response to chilling stress was also eliminated by N,N′-dimethylthiourea (DMTU, an H_2_O_2_ scavenger). In addition, ABA-induced chilling tolerance was attenuated by DMTU and diphenyleneiodonium (DPI, an H_2_O_2_ inhibitor) chloride, but AIP and AOPP had little effect on the ABA-induced mitigation of chilling stress. Na_2_WO_4_ and Flu diminished grafting- or SA-induced H_2_O_2_ biosynthesis, but DMTU and DPI did not affect ABA production induced by SA under chilling stress. These results suggest that SA participated in grafting-induced chilling tolerance by stimulating the biosynthesis of ABA and H_2_O_2_. H_2_O_2_, as a downstream signaler of ABA, mediates SA-induced chilling tolerance in grafted cucumber plants.

## 1. Introduction

Plants frequently encounter challenges from multiple abiotic stresses, such as chilling, heat, salt, drought, etc., throughout the life cycle. In all the abiotic adversity, chilling stress is considered a major limitation on crop growth and development [1]. Cucumbers (*Cucumis sativus* L.) are indigenous to tropical areas and are sensitive to chilling stress, so they often encounter low-temperature stress during winter in solar greenhouses in Northern China [2]. Therefore, the mechanism of chilling stress response and how to enhance the chilling tolerance of cucumber are currently issues that have attracted considerable attention. Grafting is the simplest and most effective technique to relieve plant stress [3]. The application of rootstocks with strong ecological adaptability can advance the resistance of grafted plants to abiotic stress [4,5], and this resistance is primarily derived from the signal interaction between rootstocks and scions. For example, grafting with heat-tolerant luffa as a rootstock can promote the heat tolerance of cucumber shoots through ABA as a long-distance signal produced by luffa roots [4]. Melatonin-induced methyl jasmonate (MeJA) and hydrogen peroxide (H_2_O_2_) accumulation serve essential roles in rootstock–scion communication and cold tolerance in watermelon [1].

Salicylic acid (SA), a plant phenolic compound with hormonal function, has been widely recognized as a signaling molecule regulating defense mechanisms in opposition to biotic and abiotic stresses [6,7]. In the early stage, SA was mainly considered a key endogenous immune signal in connection with disease resistance [8,9,10]. Recently, a great number of research results have revealed that SA is also involved in abiotic stress, such as heat, cold, salt, and drought stresses [11,12,13,14]. Additionally, many reports emphasize that SA can improve plant tolerance of adverse environmental stresses by stimulating complex signal transduction cascades [6,11]. Agnes et al. (2009) [15] showed that SA increased resistance to salt stress by promoting ABA biosynthesis. Pretreatment with SA alleviated the heat damage of maize seedlings by interacting with hydrogen sulfide (H_2_S) [16]. H_2_S is involved in the SA-induced response to chilling stress in cucumber as a downstream signal [17].

ABA, as a signaling mediator, also plays an essential role in improving the stress regulatory responses of plants [18,19]. Major abiotic stresses can induce ABA biosynthesis by promoting the activity of key biosynthetic enzymes, such as zeaxanester cyclooxygenase (ZEP) and 9-cis-epoxide carotenoid dioxygenase (NCED), and their gene expression [20,21], thus resulting in the remission of adverse environmental stresses by gene expression regulation, promoting stomatal closure and other adaptive responses [19]. Some studies indicated that exogenous ABA enhanced the response to drought and salt stresses by increasing the mRNA levels of antioxidant genes and the accumulation of osmotic adjustment substances [22,23,24]. Li et al. (2021b) [25] proved that ABA increased the cold tolerance of melon by triggering the antioxidant system.

Under various stresses, plants often activate both phytohormone signaling and reactive oxygen species (ROS, especially H_2_O_2_) [26,27], and there are interactions among these factors [28]. For instance, H_2_O_2_, as a part of ABA signaling [29], antagonistically interacts with SA in the modulation of plant stress responses [27]. A recent study involving grafting suggested that SA and ABA are involved in rootstock–scion communication, and the long-distance SA and ABA transport from the roots of rootstock to leaves of scion plays an essential part in grafting-induced cold tolerance [30,31]. ABA application enhanced cold tolerance in grafted cucumber, and this enhancement was blocked by diphenyleneiodonium chloride (DPI, a NADPH oxidase inhibitor) and N,N′-dimethylthiourea (DMTU, an H_2_O_2_ scavenger), implying that H_2_O_2_ may participate in grafting- or ABA-induced plant chilling tolerance [31]. However, whether ABA and H_2_O_2_ are involved in the SA-induced chilling stress response in grafted plants remains elusive. Therefore, we investigated the interaction of SA, ABA, and H_2_O_2_ in grafting-induced chilling response in cucumber. The present results provide compelling evidence that ABA and H_2_O_2_ function as downstream signals and participate in the SA-induced chilling tolerance of grafted cucumber. Our results provide an insight for further study on the mechanisms of rootstock–scion communication and SA signal transduction under chilling stress, which will alleviate the stress damage and enhance the adaptive capacity of cucumbers in solar greenhouses.

## 2. Results

### 2.1. Grafting Improves the Chilling Tolerance of Cucumber Plants

Here, we observed no remarkable differences in plant phenotype, photochemical efficiency (*F*_v_/*F*_m_) and actual photochemical efficiency (*Φ*_PSII_), electrolyte leakage (EL), or malondialdehyde (MDA) accumulation between grafted (*Cs*/*Cm*) and self-grafted (*Cs*/*Cs*) plants at 25 °C, except that the *Cs*/*Cm* leaves were greener than those of the *Cs*/*Cs* leaves. Chilling stress led to serious wilting; a remarkable decrease in *F*_v_/*F*_m_ and *Φ*_PSII_; and a significant increase in EL, MDA accumulation, and chilling injury index (CI) in all plants. However, the *Cs*/*Cm* plants showed mild leaf wilting; a distinctly higher *F*_v_/*F*_m_ and *Φ*_PSII_; and lower EL, MDA accumulation (Figure 1a–d), and subsequently CI than the *Cs*/*Cs* plants under chilling stress (Figure 1e).

### 2.2. Grafting Induced the Accumulation of SA, ABA, and H_2_O_2_ in Cucumbers under Chilling Stress

Chilling stress markedly increased SA accumulation (Appendix A) and phenylalanine ammonia-lyase (PAL) activity (Appendix A) in the leaves, as well as the root and xylem saps, of the grafted and self-grafted plants (*p* < 0.05); and the increases in SA content and PAL activity were distinctly greater in *Cs*/*Cm* plants than in *Cs*/*Cs* plants. These data prove that SA participates in grafting-induced chilling tolerance in cucumbers as a long-distance signal.

It is well-known that ABA is an essential signaling molecule and plays a key role in modulating plant response to abiotic stress. Therefore, we investigated whether ABA is linked to grafting-induced chilling tolerance in cucumber. At 25 °C, no marked differences were observed in ABA content, NCED activity, or *NCED*-relative mRNA expression in leaves between *Cs*/*Cs* and *Cs*/*Cm* plants (Figure 2a). However, ABA accumulation in *Cs*/*Cm* leaves increased by 150.8% following 12 h chilling stress, which was much greater than that in *Cs*/*Cs* leaves (*p* < 0.05). The NCED activity in *Cs*/*Cs* leaves changed little, but that in *Cs*/*Cm* leaves was raised by 21.6% at 12 h after chilling stress. The *NCED* mRNA abundance in the leaves of the two grafted combinations was markedly upregulated during chilling stress. The increase in *NCED* mRNA abundance was significantly greater in *Cs*/*Cm* than in *Cs*/*Cs* leaves (*p* < 0.05) following 5 °C stress for 12 h.

To explore whether H_2_O_2_ is involved in grafting-induced responses of plants to chilling stress, we determined the changes of H_2_O_2_ accumulation and *RBOH1* mRNA abundance before and after chilling stress. As you can see from Figure 2b, no remarkable differences were found in the H_2_O_2_ content and *RBOH1* mRNA abundance in leaves between *Cs*/*Cs* and *Cs*/*Cm* plants at 25 °C. After exposure to 5 °C for 6 h, the two grafted combinations showed increases in H_2_O_2_ and *RBOH1* mRNA levels in leaves. However, the *Cs*/*Cm* plants exhibited distinctly higher H_2_O_2_ accumulation and *RBOH1* mRNA abundance in response to chilling stress than the *Cs*/*Cs* plants.

### 2.3. SA Improves Grafting-Induced Chilling Tolerance and Accumulation of ABA and H_2_O_2_ in Cucumber Plants

Figure 3a shows that exogenous SA markedly decreased, while AIP and AOPP increased the chilling damage in grafted plants, compared with the deionized water (H_2_O)-treated plants (*p* < 0.05), after chilling treatment for 48 h. SA also reduced the EL and CI in self-grafted and grafted cucumbers under chilling stress. However, AIP and AOPP increased the EL and CI relative to the H_2_O treatment (*p* < 0.05) under chilling stress. *Cs*/*Cm* plants showed a noticeably lower EL and CI (*p* < 0.05) in all treatments than *Cs*/*Cs* plants during chilling (Figure 3b,c).The mRNA abundances of *CsCBF1* and *CsCOR47* in SA-treated *Cs*/*Cs* leaves increased 6.58-fold and 7.04-fold, respectively, and those in SA-treated *Cs*/*Cm* leaves increased 8.39-fold and 8.55-fold, respectively, under chilling stress (Figure 3d,e); the levels were much higher than those in H_2_O-treated *Cs*/*Cs* and *Cs*/*Cm* leaves (*p* < 0.05). However, AIP- and AOPP-treated *Cs*/*Cs* and *Cs*/*Cm* leaves exhibited significantly decreased mRNA abundances of *CsCBF1* and *CsCOR47* relative to the H_2_O treatments at 48 h after chilling exposure (*p* < 0.05).

Interestingly, SA dramatically increased the *NCED* mRNA abundance and ABA content in *Cs*/*Cs* leaves at 25 °C (Figure 4a,b). After exposure to chilling stress for 24 h, the ABA accumulation in the H_2_O-treated *Cs*/*Cs* and *Cs*/*Cm* plants was increased 1.97-fold and 2.30-fold, respectively, whereas that in SA-treated plants of *Cs*/*Cs* and *Cs*/*Cm* increased by 2.92-fold and 3.34-fold, respectively, compared with the control. AIP and AOPP treatments exhibited little difference in ABA content relative to the H_2_O-treated plants during chilling stress, but the increase in ABA content was much greater in *Cs*/*Cm* plants than in *Cs*/*Cs* plants (Figure 4e). Similarly, at 25 °C, the SA treatment showed marked increases in *RBOH1* mRNA abundance and H_2_O_2_ content in *Cs*/*Cs* leaves (Figure 4c,d). After the plants were exposed to chilling for 24 h, the H_2_O_2_ accumulation in the SA-treated *Cs*/*Cs* and *Cs*/*Cm* leaves increased by 123.1% and 166.1%*,* respectively, and was great higher than that in the H_2_O-treated *Cs*/*Cs* and *Cs*/*Cm* leaves (*p* < 0.05). No significant differences were observed in H_2_O_2_ accumulation among AIP-, AOPP-, and H_2_O-treated plants under chilling stress; nonetheless, a noticeable increase in H_2_O_2_ content was detected in *Cs*/*Cm* plants relative to *Cs*/*Cs* plants (Figure 4f).

### 2.4. The Roles of ABA and H_2_O_2_ in SA- or Grafting-Induced Chilling Tolerance in Cucumber

To validate the roles of ABA and H_2_O_2_ in chilling tolerance induced by SA or grafting, we first detected the effect of exogenous ABA and its synthetic inhibitors NaWO_4_ and Flu on the EL and CI in *Cs*/*Cs* and *Cs*/*Cm* plants. As shown in Figure 5, the increases in EL and CI caused by chilling were greatly reduced by 50 μM ABA and 1 mM H_2_O_2_, but enhanced by 3 mM Na_2_WO_4_ and 50 μM Flu or 5.0 mM DMTU and 0.1 mM DPI. The *Cs*/*Cm* plants exhibited much lower EL and CI values than the *Cs*/*Cs* plants (*p* < 0.05). Moreover, Na_2_WO_4_ or DMTU also weakened the SA-induced decrease in EL and CI after exposure to chilling stress. These data indicate that ABA and H_2_O_2_ are involved in SA- or grafting-induced chilling tolerance.

### 2.5. The Role of ABA in Grafting-Induced Biosynthesis of SA and H_2_O_2_ in Cucumber Leaves

To explore the interaction between ABA and SA or H_2_O_2_ signals, we first measured the effects of ABA on biosynthesis and chilling tolerance in *Cs*/*Cs* or *Cs*/*Cm* plants. Figure 6a shows that no outstanding differences were found in *PAL* mRNA abundance or SA content between ABA- and H_2_O-treated *Cs*/*Cs* plants at 25 °C. After being exposed to chilling for 24 h, all the ABA-, Na_2_WO_4_-, Flu-, and H_2_O-treated plants exhibited a distinct increase in the SA content and had no marked differences among the treatments (Figure 6c). However, the increase in SA content caused by chilling stress was extremely greater in *Cs*/*Cm* than in *Cs*/*Cs* leaves (*p* < 0.05). Furthermore, the *Cs*/*Cm* plants showed lower EL and CI, but higher *F*_v_/*F*_m_ than the *Cs*/*Cs* plants (*p* < 0.05); however, AIP and AOPP did not affect the ABA-induced decrease in EL and CI or the increase in *F*_v_/*F*_m_ in leaves of *Cs*/*Cs* or *Cs*/*Cm* plants during chilling treatment (Figure 6d–f).

We also determined the response of H_2_O_2_ biosynthesis to ABA in *Cs*/*Cs* or *Cs*/*Cm* plants. The application of ABA markedly increased the *RBOH1* mRNA abundance and H_2_O_2_ accumulation in the leaves of *Cs*/*Cs* and *Cs*/*Cm* plants after exposure to chilling stress (Figure 7a,b), but Na_2_WO_4_- and Flu attenuated the ABA- and grafting-induced production of under chilling stress (Figure 7c). Moreover, DMTU and DPI treatments abolished ABA-induced chilling tolerance, as evidenced by higher EL and CI, but lower *F*_v_/*F*_m_ (Figure 7d–f). Thus, we consider that ABA and H_2_O_2_ participate in SA-induced chilling tolerance in grafted cucumber.

### 2.6. Interaction of ABA and H_2_O_2_ in SA- or Grafting-Induced Chilling Tolerance in Cucumber Plants

To further explore the upstream and downstream relationships between ABA and H_2_O_2_ in the SA-mediated chilling stress response in grafted plants, we investigated the effects of Na_2_WO_4_ and Flu on SA-induced H_2_O_2_ biosynthesis and those of DMTU and DPI on SA-induced ABA production. SA significantly increased the *RBOH1* mRNA level and H_2_O_2_ content, and the increases in *RBOH1* mRNA abundance and H_2_O_2_ content in *Cs*/*Cm* leaves were much higher than those in *Cs*/*Cs* leaves. Na_2_WO_4_ and Flu completely blocked the SA- or grafting-induced increase in *RBOH1* mRNA abundance and H_2_O_2_ accumulation (Figure 8), indicating that ABA is involved in SA- and grafting-induced H_2_O_2_ synthesis. SA also increased the *NCED* mRNA abundance and ABA content, especially in *Cs*/*Cm* leaves. However, DMTU and DPI exhibited little effect on SA-induced ABA production in self-grafted and grafted cucumber plants (Figure 9). These results confirmed that H_2_O_2_ is located downstream of ABA and participates in SA-induced chilling tolerance in grafted cucumber plants.

## 3. Discussion

### 3.1. SA Is Involved in Grafting-Induced Chilling Tolerance

As we all know, hetero-rooted grafting can enhance plant resistance to abiotic stresses, such as heat, chilling, salinity, drought, etc. [1,4,32,33]. Here, we found that *Cs*/*Cm* plants exhibited mild chilling-damage symptoms; higher *F_v_/F_m_* and *Φ*_PSII_; and lower EL, MDA accumulation, and CI relative to *Cs*/*Cs* under chilling stress (Figure 1). These data are consistent with our previous results [30]. A great deal of evidence has indicated that SA can enhance chilling tolerance in various plant species [17,34], and this enhancement has something to do with the upregulation of cold-responsive (COR) gene expression [30]. In this study, we observed that SA accumulation was significantly higher in *Cs*/*Cm* than in *Cs*/*Cs* during chilling stress (Appendix A), and this observation is in agreement with the previous studies [30]. Exogenous SA improved the chilling tolerance of the two grafted combination plants, but application of the SA inhibitor AIP or AOPP reduced grafting-induced chilling tolerance (Figure 3). These results indicate that SA serves a crucial role in grafting-induced chilling tolerance. The increase in SA accumulation was not only in leaves, but also in the roots and xylem of *Cs*/*Cm* after the plants were treated at 5 °C stress. These data further confirm that SA can be transported from rootstock roots to scion leaves through xylem. SA, as a long-distance signaler, is involved in rootstock-induced chilling resistance in cucumber plants.

### 3.2. SA and ABA Function Together in Grafting-Induced Chilling Tolerance

It is well documented that ABA, as a central regulator, plays essential roles in plant response to chilling stress [25,35,36]. Lv et al. (2022) [31] reported that chilling stress upregulated the mRNA expression of *NCED* to increase ABA accumulation, and the increase in ABA content was proportional to the chilling tolerance of cucumber varieties. The present results exhibited that grafting-improved ABA accumulation, NCED activity, and mRNA level of *NCED* in cucumber leaves under chilling stress (Figure 1a). Exogenous ABA increased the chilling tolerance, but the inhibitors of ABA biosynthesis Na_2_WO_4_ and Flu reduced the chilling tolerance of grafted plants (Figure 5a), indicating that ABA is involved in the grafting-induced chilling tolerance of cucumber.

Some studies proved the crosstalk between SA and ABA in plants resistant to abiotic stress. For example, Wang et al. (2002) [37] found that, after heat acclimation or external application of SA, grape seedlings showed similar changes in endogenous ABA content between the two treatments, and the rate-limiting enzyme of SA synthesis, PAL, also showed similar change trends. Therefore, the authors speculated that SA might induce heat tolerance in grapes through the ABA signaling pathway. Under drought stress, both SA and ABA can regulate the mRNA expression of DREB in tobacco, indicating that an interaction between them in the signal transduction process [38]. Shakirova et al. (2016) [39] also verified that SA can induce endogenous ABA accumulation in wheat under stress conditions and serve an important role in alleviating cadmium stress. Here, we observed that SA markedly increased the *NCED* mRNA abundance and ABA accumulation (Figure 4a,b), while ABA did not affect endogenous SA synthesis in *Cs*/*Cs* leaves at room temperature (Figure 6a,b). AIP or AOPP reduced the chilling- and grafting-induced ABA accumulation in the leaves of grafted plants, but the pretreatment with Na_2_WO_4_ or Flu did not affect the chilling- and rootstock-induced increase in SA (Figure 4e and Figure 6c). Moreover, SA-induced chilling tolerance was attenuated by Na_2_WO_4_ and Flu, whereas AIP and AOPP showed little effect on ABA-induced resistance to chilling stress (Figure 5c and Figure 6d–f). These results reveal that ABA, as a downstream signal, interacting with SA, participates in the grafting-induced plant response to chilling stress.

### 3.3. ABA Mediates SA- and Grafting-Induced H_2_O_2_ Accumulation and Chilling Tolerance

Increasing evidence suggests that H_2_O_2_, as an important second messenger, has a crucial role in plant responses to abiotic stresses [1,40,41]. For example, Sun et al. (2019) [42] revealed that H_2_O_2_, on the one hand, improved the heat tolerance of tomato plants and, on the other hand, regulated antioxidant enzyme activities to control the total H_2_O_2_ at a level that is beneficial to heat stress memory, and maintaining a lower H_2_O_2_ level to address future challenges against heat stress. Liu et al. (2020) [43] found that H_2_O_2_ increased chilling tolerance by upregulating the mRNA levels of cold-responsive genes and improving CO_2_ assimilation and photoprotection in cucumber. Our present study revealed that grafting with pumpkin as a rootstock increased H_2_O_2_ accumulation and *RBOH1* mRNA expression under chilling stress (Figure 2b). Exogenous H_2_O_2_ improved the chilling tolerance of grafted cucumber plants, but DMTU and DPI markedly reduced grafting-induced chilling tolerance (Figure 5b). These indicate that H_2_O_2_ is involved in grafting-induced chilling tolerance, in accordance with the result of Li et al. (2021a) [1] found in grafted watermelon.

Previous studies have shown that SA is positively or negatively associated with H_2_O_2_ in plant response to stress conditions [44]. In the current study, we found that SA increased the *RBOH1* mRNA abundance and H_2_O_2_ accumulation in *Cs*/*Cs* leaves (Figure 4c,d), but no significant differences were observed in either *PAL* mRNA level or SA content between H_2_O_2_- or H_2_O-treated plant leaves (Appendix A). AIP and AOPP repressed the chilling-induced H_2_O_2_ accumulation in the leaf tissues of grafted plants (Figure 4f). In addition, DMTU weakened or blocked the SA-induced chilling tolerance of grafted plants (Figure 5c). These data indicate that H_2_O_2_, as a downstream signal, participates in the SA-induced chilling tolerance of grafted cucumber.

H_2_O_2_ is an important second messenger in ABA signaling in guard cells [45]. Hu et al. (2005) [46] indicated that H_2_O_2_, as a crucial signaler, plays a vital role in ABA-induced stomatal closure. Li et al. (2018) [47] showed that H_2_O_2_ is involved in the ABA-induced development of adventitious roots under drought stress by regulating the proteins relative to photosynthesis, stress defense, folding, modification, and degradation in tomato plants. Recently, some studies revealed crosstalk between H_2_O_2_ and ABA in the defense against abiotic stresses of grafted plants. For instance, Li et al. (2014a) [4] showed that luffa rootstock increased the heat tolerance of grafted cucumber, which was attributed to ABA accumulation in rootstock roots caused by high temperature. The accumulated ABA in rootstock roots was transferred to scion leaves and triggered the expression of the heat shock protein HSP70 through an apoplast-H_2_O_2_-dependent manner. Lv et al. (2022) [31] reported that H_2_O_2_ was involved in the grafting-induced cold tolerance of grafted cucumber caused by ABA. The present data exhibited that ABA obviously stimulated the synthesis of H_2_O_2_ (Figure 7a, b), whereas H_2_O_2_ exerted little effect on ABA generation in *Cs*/*Cs* plants (Appendix A). Application of Na_2_WO_4_ and Flu markedly prevented ABA- or grafting-induced H_2_O_2_ accumulation in grafted plants under chilling stress (Figure 7c). DMTU and DPI attenuated the ABA-induced chilling tolerance of self-grafted and grafted plants (Figure 7d–f). These data indicate that ABA mediates the SA-induced chilling tolerance of grafted cucumber by activating H_2_O_2_ biosynthesis and accumulation. Moreover, Na_2_WO_4_ and Flu markedly decreased SA-induced *RBOH1* mRNA expression and H_2_O_2_ accumulation (Figure 8), while DMTU and DPI exhibited no significant effect on SA-induced *NCED* and ABA levels (Figure 9) in grafted cucumber plants. Therefore, we suggest that H_2_O_2_ was located downstream of ABA and involved in SA-induced chilling tolerance of grafted cucumber.

## 4. Materials and Methods

### 4.1. Plant Materials and Growth Conditions

In the present study, the pumpkin (*Cucurbita moschata* D., *cv*. Jinmama 519, *Cm*) and cucumber (*Cucumis sativus* L., *cv*. ‘Jinyou 35’, *Cs*) were used as rootstock and scion, respectively. Germinated seeds for rootstock were sown in nutritive bowls filled with substrate, in accordance with our previous study [30], and seeds for scion were sown after 3 d. When the cotyledon of the rootstock was fully expanded and that of scion was initially unfolded, grafting was carried out by using the top insertion grafting method. The resulting grafted and self-grafted seedlings, named *Cs*/*Cm* and *Cs*/*Cs*, respectively, were grown in an artificial climate chamber, with a photoperiod of 12 h/12 h (day/night), temperature of 26/18 °C (day/night), humidity of 90–95%, and photon flux density (PFD) of 50 μmol m^−2^∙s^−1^. About 5–7 d later, the environmental conditions of climate chamber were regulated to 600 μmol m^−2^∙s^−1^ PFD, 26/18 °C (day/night) temperature, 11 h photoperiod, and 70~80% relative humidity.

### 4.2. Experimental Design

#### 4.2.1. Effect of Grafting on the Chilling Tolerance of Cucumber

The self-grafted (*Cs*/*Cs*, control) and heterografted (*Cs*/*Cm*) plants with three true leaves were transferred into growth chambers at 5 °C for chilling stress treatment or 25 °C for the control treatment. Twenty-four hours later, the samples were collected to measure the maximum photochemical efficiency (*F*_v_/*F*_m_), actual photochemical efficiency (*Φ*_PSII_), electrolyte leakage (EL), and malondialdehyde (MDA) content. The chilling injury index (CI) of *Cs*/*Cs* and *Cs*/*Cm* was also determined after treatment for 72 h. The accumulation levels of SA, ABA, and H_2_O_2_, as well as their key biosynthesis enzyme activities and/or gene expression levels, were measured following 12 h of chilling stress. Three biological replicates with 20 plants per replicate were performed for each treatment.

#### 4.2.2. SA, ABA, and H_2_O_2_ Interaction Are Involved in Grafting-Induced Chilling Tolerance of Cucumber Plants

At the three true leaf stage, the *Cs*/*Cs* and/or *Cs*/*Cm* leaves were foliar sprayed with 1.0 mM SA, 50 μM ABA, 1 mM H_2_O_2_ [48,49,50], or deionized water (control). Twenty-four hours later, the plants were exposed to 8 °C/5 °C (day/night). The plant phenotypes, EL, CI, MDA content, *F*_v_/*F*_m_ and *Φ*_PSII_, and mRNA abundances of *CBF1* and *COR47* were determined after plants were exposed to chilling for 12 h, 48 h, or 72 h. To suppress the production of SA, ABA, or H_2_O_2_, the *Cs*/*Cs* and *Cs*/*Cm* plants were pretreated with 30 μM 2-aminoindan-2-phosphonic acid (AIP), 0.1 mM L-a-aminooxy-b-phenylpropionic acid (AOPP) (biosynthesis inhibitors of SA), 3 mM sodium tungstate (Na_2_WO_4_), 50 μM fluridone (Flu) (biosynthesis inhibitors of ABA), 5 mM DMTU (a scavenger of H_2_O_2_), or 0.1 mM DPI (biosynthesis inhibitors of H_2_O_2_). At 12 h after being pretreated with the inhibitors or scavenger, the plants were foliar sprayed with SA or ABA, and 12 h later, they were exposed to chilling. To distinguish from the control, the deionized water treatment after chilling stress was designated as H_2_O treatment. *Cs*/*Cs* plants were pretreated with 1 mM SA, 50 μM ABA, 1 mM H_2_O_2_, or deionized water (control) to analyze the effects of SA on endogenous ABA and H_2_O_2_ biosynthesis, of ABA on endogenous H_2_O_2_ and SA biosynthesis, and of H_2_O_2_ on ABA and SA production. Three biological replicates with 15 plants per replicate were performed for each treatment.

#### 4.2.3. Effect of ABA Inhibitor on SA-Induced H_2_O_2_ Synthesis in Grafted Cucumber

The plants at the three true leaf stage were pretreated with 1.0 mM SA, 3 mM Na_2_WO_4_, 50 μM Flu, 3 mM Na_2_WO_4_ + 1.0 mM SA, 50 μM Flu + 1.0 mM SA, or deionized water (H_2_O). Twelve hours later, the pretreated plants were exposed to 5 °C. At 12 h after chilling treatment, the respiratory burst oxidase homolog (*RBOH1*) mRNA abundance and H_2_O_2_ accumulation were measured. Three biological replicates with 10 plants per replicate were performed for each treatment.

#### 4.2.4. Effect of Scavenger and Inhibitor of H_2_O_2_ on SA-Induced ABA Synthesis in Grafted Cucumber

The plants at the three true leaf stage were pretreated with 1.0 mM SA, 5.0 mM DMTU, 0.1 mM DPI, 5.0 mM DMTU + 1.0 mM SA, 0.1 mM DPI + 1.0 mM SA, or deionized water (H_2_O). At 12 h after pretreatment, the plants were exposed to 5 °C for 12 h to determine the *NCED* mRNA abundance and ABA accumulation. Three biological replicates with 10 plants per replicate were performed for each treatment.

### 4.3. Measurement of CI, EL, and MDA

The chilling-stressed plants were graded by following the method of Semeniuk et al. (1986) [51]. Grade 1: The margin of the first or second leaf from the bottom shows yellow or is slightly wilting, but no damage symptom in the upper leaves. Grade 2: The margin of the first and second leaves from the bottom is dehydrated severely, and the third leaf is yellow or slightly wilting, but the heart leaf has no obvious cold-damage symptom. Grade 3: The first and second leaves from the bottom emerge with dehydration spots. The margin of third leaf shows severe dehydration, and the heart leaf is slightly wilting. Grade 4: The dehydrating spots on the first and second leaves from the bottom are joined together, and the third leaf shows dehydration spots. The heart leaf is markedly wilting, but it can still recover at room temperature. Grade 5: All the leaves are withered and cannot be recovered at normal temperature. The CI was calculated by using the following formula: CI = Σ (plants of different grade × grade)/[total plants × 5 (the maximum grade)]. EL was assayed by using the method of Dong et al. (2013) [52]. The 0.3 g leaf tissues were submerged in 25 °C deionized water for 3 h, and the initial electrical conductivity (EC1) was detected with a conductivity meter (DDB-303A, Shanghai, China). Then the leaf tissues were held in boiling water for 30 min, and the final electrical conductivity (EC2) was detected after cooling to normal temperature. EL was calculated as EC1/EC2. MDA content was determined by Heath and Packer’s method (1968) [53].

### 4.4. Detection of F_v_/F_m_ and φ_PSII_

The *F*_v_/*F*_m_ and *φ*_PSII_ were measured with a portable pulse-modulated fluorometer (FMS-2, Hansatech, Norfolk, UK), as described by Bi et al. (2017) [54]. The *F*_v_/*F*_m_ and *φ*_PSII_ values were calculated as follows: *F*_v_/*F*_m_ = (*F*_m_ − *F*_0_)/*F*_m_′; *Φ*_PSII_ = (*F*_m_′ − *F*_s_)/*F*_m_′ (Demmig-Adams and Adams 1996) [55]. The visualization of the *F*_v_/*F*_m_ and *φ*_PSII_ imaging was performed by using a chlorophyll fluorescence imaging system (Imaging PAM, FluorCam, Czech Republic), as described by Tian et al. (2017) [56].

### 4.5. SA Content and PAL Activity Assay

The SA content was assayed with HPLC–MS (Thermo Fisher Scientific, TSQ Quantum Access, USA), following the method of Li et al. (2014b) [57], but with minor modification by Fu et al. (2021) [30], and it was calculated according to the standard curve, employing an SA standard sample (Sigma-Aldrich, Burlington, MA, USA). The PAL activity was measured with a spectrophotometer (UV-2450, Shimadzu, Japan), as described by Fu et al. (2021) [30]. In brief, the leaf sample was homogenized in 5 mL of 50 mM Tris–HCl buffer (pH 8.5) containing 5 mM EDTA, 15 mM β-mercaptoethanol, and 0.15% (*w*/*v*) polyvinylpyrrolidone. After the homogenate was centrifuged at 14,000× *g* for 20 min at 4 °C, the resulting supernatant was added into 0.02 M phenylalanine and reacted at 37 °C for 1 h. The PAL activity was expressed as the enzyme amount required for the changes in absorbance of 0.01 h^−1^ at 290 nm.

### 4.6. Determination of ABA and H_2_O_2_ Contents

The ABA content was detected by using HPLC–MS (Thermo Fisher Scientific, TSQ Quantum Access, Waltham, MA, USA), as described by Li et al. (2014b) [57], with minor modification by Lv et al. (2022) [31]. Briefly, 0.3 g of frozen leaf tissues was ultrasonically extracted in 5 mL of 80% methanol extraction medium (containing 30 μg·mL^−1^ sodium diethyldithiocarbamate) for 20 min and then placed under −20 °C in the dark for 16 h. Then the extract solution was centrifugated at 3200× *g* for 10 min at 4 °C, and the supernatant was dried by using reduction vaporization at 38 °C. After pigments and phenolics were removed with chloroform and polyvinylpyrrolidone (PVP), respectively, 3.5 mL supernatant was mixed with 3 mL ethyl acetate, oscillated for 15 min, and then left to stand for 10 min. The ester phase was collected and evaporated to dryness under reduced pressure at 36 °C; then it was dissolved in a 1.0 mL methanol–acetic acid solution (45:55, v:v, acetic acid concentration 0.04%). Finally, the ester phase was filtered with a 0.2 µm filter membrane for LC/MS analysis. The chromatographic conditions were the same as those described by Lv et al. (2022) [31]. We constructed a standard curve by using an ABA standard sample (Sigma-Aldrich, Burlington, MA, USA) and calculated the ABA content according to the peak area. H_2_O_2_ accumulation was quantified by using an H_2_O_2_ kit (A064-1, Nanjing Jiancheng Bioengineering Institute, Nanjing, China). Simultaneously, H_2_O_2_ fluorescence imaging was also visualized at the subcellular level with the H_2_O_2_-sensitive fluorescent probe 2′,7′-dichlorodihydrofluorescein diacetate (H2DCFDA) (MCE, Cat. No. HY-D0940, Shanghai, China), as described by Zhang et al. (2020) [58].

### 4.7. Quantitative Real-Time PCR Analysis

RNAs from *Cs*/*Cs* and *Cs*/*Cm* leaves were extracted by using TRIzol RNA separation reagent (TRANs, Beijing, China). The reverse transcription was carried out with HiScript^®^ III RT SuperMix (Vazyme, Nanjing, China). The mRNA abundances of *NCED, RBOH1, CBF1, COR47*, and *PAL* in plants were analyzed by qRT-PCR, using a qRT-PCR system (Vazyme, Nanjing, China). To assess the normalization degree, the cucumber β-action gene (Solyc11g005330) was used as the reference gene. The primers for qRT-PCR are shown in Appendix A.

### 4.8. Xylem Sap and Roots Collection

Xylem sap was collected by following the method of Li et al. (2014a) [4], with minor modification. Cut the stem 3 cm above the junction of scion and rootstock, using a blade, and cover the incision immediately with a centrifuge tube containing absorbent cotton to collect xylem exudate. Then squeeze the absorbent cotton and suck the xylem sap out with a syringe. The roots of the self-grafted and grafted plants were cut with scissors, placed in a triangle, and then dried with a vacuum drier for 3 days. The collected sap and dried root samples were stored at −80 °C for analysis of the SA content and PAL activity.

### 4.9. Accession Numbers

*CsActin* (XM_011659465), *CmActin* (XM_023107141), *CsCBF1* (XM_004140746), *CsCOR* (XM_011659051), *PAL* (XM_011659349), *NCED* (XM_004147720), and *RBOH1* (XM_004135565).

## 5. Conclusions

In summary, the current study demonstrates that pumpkin as rootstock quickly accumulates SA in the roots under chilling stress and then transports it to the scion leaves and simultaneously induces the biosynthesis of SA in the scion leaves. The rootstock-induced SA accumulation in leaves promotes ABA production and further stimulates H_2_O_2_ accumulation, and this subsequently activates the ICE-CBF-COR signal to mitigate the chilling damage of membrane lipid peroxidation and photosynthetic apparatus (Figure 10), thus finally improving the *Cs*/*Cm* plant tolerance against chilling stress. Therefore, the SA-induced accumulation of ABA and subsequent H_2_O_2_ are involved in rootstock–scion communication and play a crucial role in the grafting-induced response to chilling stress. To better elucidate the molecular mechanism of SA and how H_2_O_2_ mediates SA-induced chilling tolerance in grafted plants, further research using transgenic plants or mutant and advanced molecular techniques analyses is required.

## Figures and Tables

**Figure 1 ijms-23-16057-f001:**
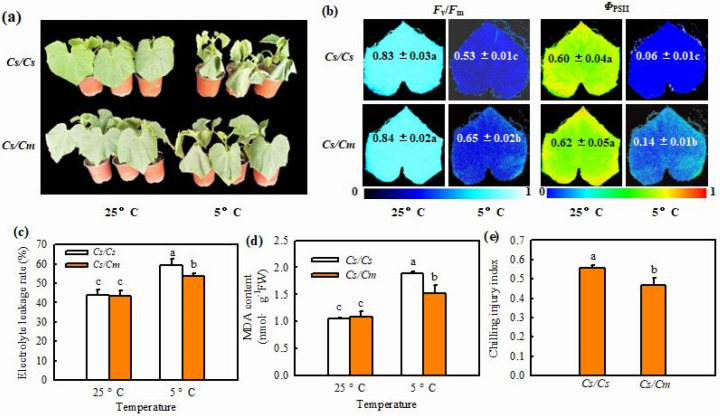
Effect of grafting on the chilling tolerance of cucumber. (**a**) Plant phenotype. The *Cs*/*Cs* and *Cs*/*Cm* plants were treated at 5 °C for 0 d and 48 h. (**b**) Images of *F*_v_/*F*_m_ and *Φ*_PSII_. The false color coding from 0 to 1.0 depicted at the bottom of image indicates the degree of photoinhibition at PSII. (**c**) Electrolyte leakage. (**d**) MDA content. The *Cs*/*Cs* and *Cs*/*Cm* plants were treated at 5 °C for 0 h and 24 h. (**e**) Chilling injury index. The *Cs*/*Cs* and *Cs*/*Cm* plants were treated at 5 °C for 72 h. Data are presented as the means ± SD (*n* = 3). Different letters indicate a statistical significance between samples at *p* < 0.05.

**Figure 2 ijms-23-16057-f002:**
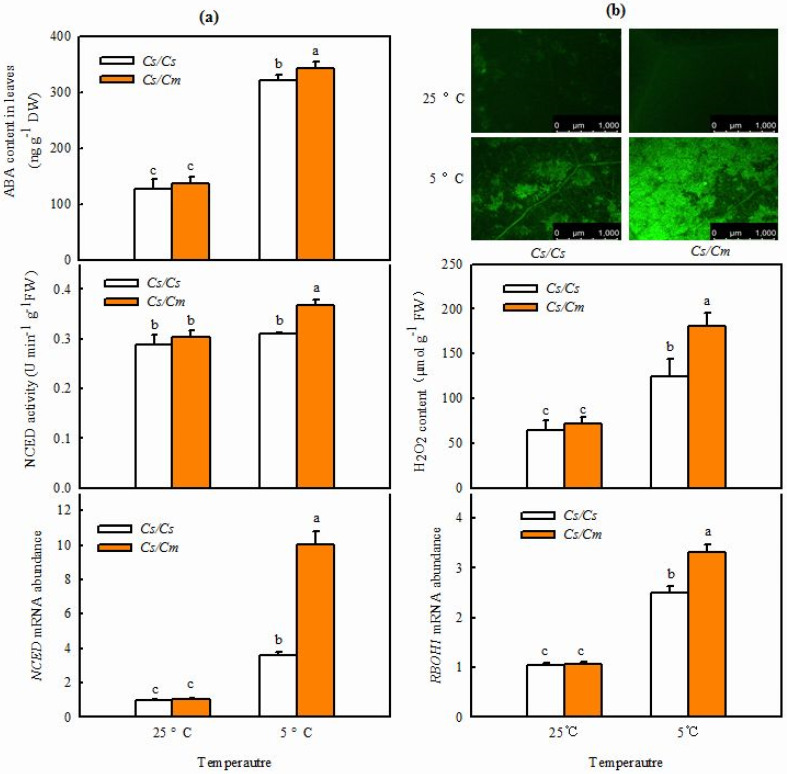
Accumulation of ABA and H_2_O_2_ in leaves of grafted or self−grafted cucumber plants. (**a**) ABA content, NCED activity, and its mRNA abundance, respectively. The plants were treated at 5 °C for 0 h and 12 h. (**b**) H_2_O_2_ accumulation and *RBOH1* mRNA abundance, respectively. The plants were treated at 5 °C for 0 h and 6 h. Data are presented as the means ± SD (*n* = 3). Different letters indicate a statistical significance between samples at *p* < 0.05.

**Figure 3 ijms-23-16057-f003:**
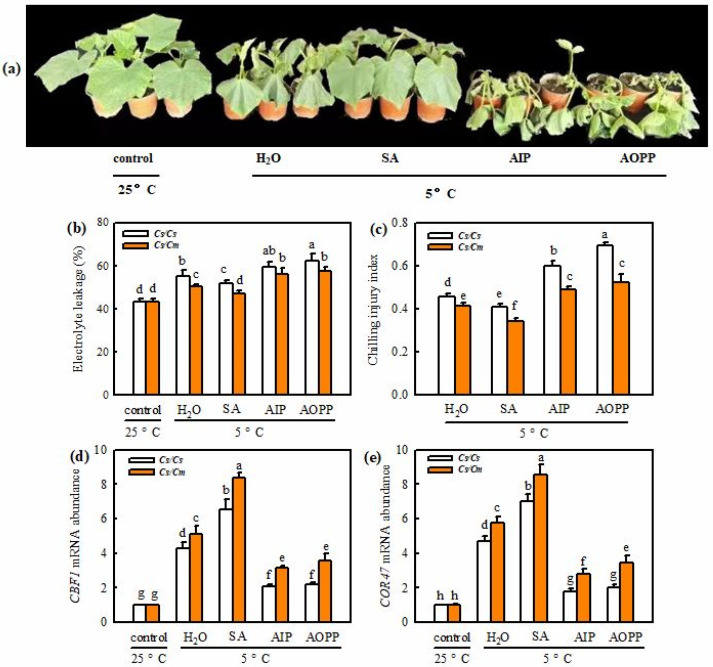
Involvement of SA in grafting-induced chilling tolerance. Plants with three leaves were pretreated with 1.0 mM SA, 30 μM AIP, 0.1 mM AOPP, or deionized water (H_2_O, control) for 24 h and then exposed to chilling stress. (**a**) Phenotype of *Cs*/*Cm* plants after chilling treatment for 0 h and 72 h. (**b**) Electrolyte leakage of the *Cs*/*Cs* and *Cs*/*Cm* plants after being treated at 5 °C for 0 h and 48 h. (**c**) Chilling injury index of *Cs*/*Cs* and *Cs*/*Cm* plants treated at 5 °C for 72 h. (**d**,**e**) Relative mRNA expression of *CBF1* and *COR47* of *Cs*/*Cs* and *Cs*/*Cm* plants treated at 5 °C for 0 h and 24 h. Data are presented as the means ± SD (*n* = 3). Different letters indicate a statistical significance between samples at *p* < 0.05.

**Figure 4 ijms-23-16057-f004:**
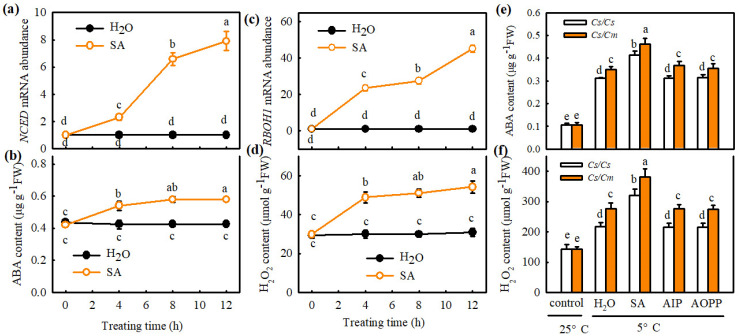
SA is involved in grafting-induced accumulation of ABA and H_2_O_2_ in cucumber leaves. (**a**,**b**) Changes of *NCED* mRNA abundance and ABA content in *Cs*/*Cs*, respectively, at 25 °C. (**c**,**d**) Changes of *RBOH1* mRNA abundance and H_2_O_2_ content in *Cs*/*Cs*, respectively, at 25 °C. Plants with three leaves were pretreated with 1.0 mM SA or distilled water (control) for 12 h. (**e**,**f**) ABA and H_2_O_2_ contents in *Cs*/*Cs* and *Cs*/*Cm* leaves, respectively, before and after exposure 5 °C for 24 h. Plants were pretreated with 1.0 mM SA, 30 μM AIP, 0.1 mM AOPP (two biosynthetic inhibitors of SA), or deionized water (control), respectively, for 12 h and then exposed to chilling stress. Data are presented as the means ± SD (*n* = 3). Different letters indicate a statistical significance between samples at *p* < 0.05.

**Figure 5 ijms-23-16057-f005:**
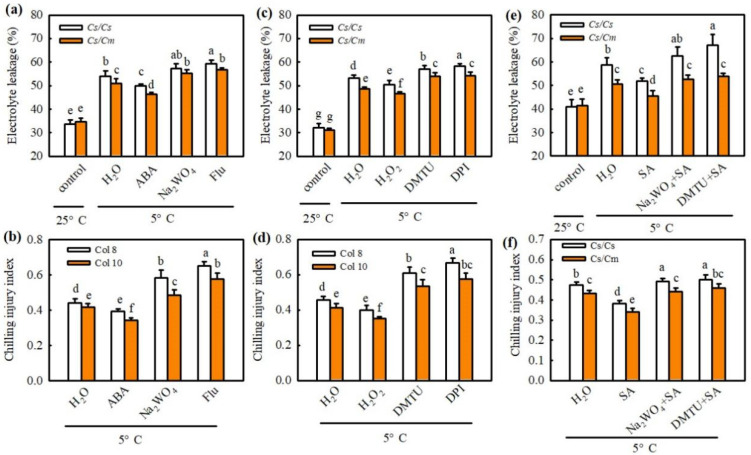
The role of ABA and H_2_O_2_ in SA- or grafting-induced chilling tolerance in cucumber. (**a**) Electrolyte leakage of plants after chilling treatment for 0 h and 48 h. (**b**) Chilling injury index of plants after chilling treatment for 0 h and 72 h. Plants were foliar sprayed with 50 μM ABA, 3 mM Na_2_WO_4_, 50 μM fluridone, or distilled water (control) for 12 h and then exposed to chilling stress. (**c**) Electrolyte leakage of plants after chilling treatment for 0 h and 48 h. (**d**) Chilling injury index of plants after chilling treatment for 0 h and 72 h. Plants were foliar sprayed with 1 mM H_2_O_2_, 5 mM DMTU, 0.1 mM DPI, or distilled water (control) for 12 h and then exposed to chilling stress. (**e**) Electrolyte leakage of plants after chilling treatment for 0 h and 48 h or 72 h, respectively. (**f**) Chilling injury index of plants after chilling treatment for 0 h and 72 h. Plants were pretreated with 1 mM SA, 3 mM Na_2_WO_4_ + 1 mM SA, 5 mM DMTU + 1 mM SA, or distilled water (control) for 12 h and then exposed to chilling stress. Data are presented as the means ± SD (*n* = 3). Different letters indicate a statistical significance between samples at *p* < 0.05.

**Figure 6 ijms-23-16057-f006:**
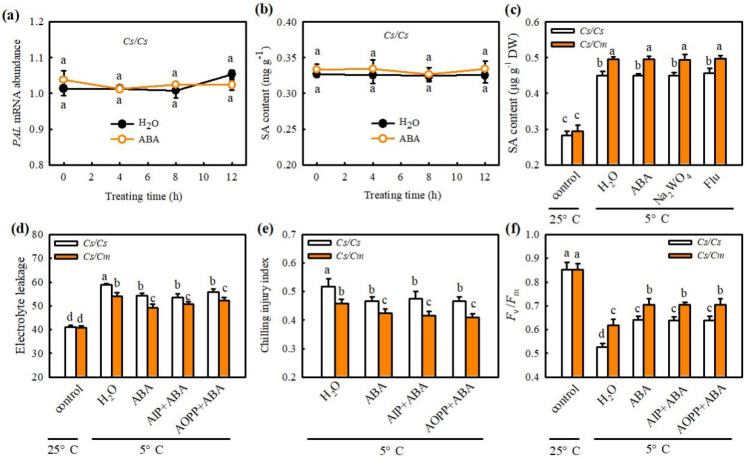
Effect of ABA on the SA biosynthesis and the role of SA− in ABA−induced chilling tolerance in grafted cucumber. (**a**,**b**) *PAL* mRNA abundance and SA content, respectively, in *Cs*/*Cs* leaves. The *Cs*/*Cs* cucumber were pretreated with 50 μM ABA or distilled water (control), and then the changes of *PAL* mRNA abundance and SA content were measured within 12 h. (**c**) SA accumulation in *Cs*/*Cs* or *Cs*/*Cm* after chilling treatment for 0 h and 12 h. Plants were foliar sprayed with 50 μM ABA, 3 mM Na_2_WO_4_, 50 μM Flu, or distilled water (control) for 12 h and then exposed to chilling stress. (**d**) Electrolyte leakage of grafted plants after chilling treatment for 0 h and 48 h. (**e**) Chilling injury index of grafted plants before and after chilling treatment for 72 h. (**f**) *F*_v_/*F*_m_ of *Cs*/*Cs* or *Cs*/*Cm* plants after chilling treatment for 0 h and 48 h. Plants were treated with 50 μM ABA, 30 μM AIP + 50 μM ABA, 0.1 mM AOPP + 50 μM ABA, or distilled water (control) for 12 h and then exposed to chilling stress. Data are presented as the means ± SD (*n* = 3). Different letters indicate a statistical significance between samples at *p* < 0.05.

**Figure 7 ijms-23-16057-f007:**
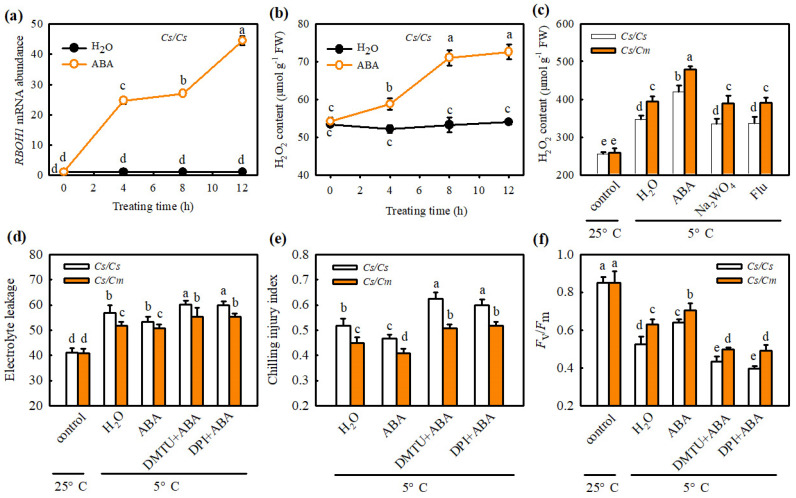
Effect of ABA on the H_2_O_2_ biosynthesis and the role of H_2_O_2_ in ABA−induced chilling tolerance in grafted cucumber. (**a**,**b**) *RBOH1* mRNA abundance and H_2_O_2_ accumulation, respectively, in *Cs*/*Cs* leaves. The *Cs*/*Cs* plants were pretreated with 50 μM ABA or distilled water (control), and then we measured the changes of *RBOH1* mRNA abundance and H_2_O_2_ content within 12 h. (**c**) H_2_O_2_ accumulation in *Cs*/*Cs* or *Cs*/*Cm* after chilling treatment for 0 h and 12 h. Plants were foliar sprayed with 50 μM ABA, 3 mM Na_2_WO_4_, 50 μM Flu, or distilled water (control) for 12 h and then exposed to chilling stress. (**d**) Electrolyte leakage of *Cs*/*Cs* or *Cs*/*Cm* plants after chilling treatment for 0 h and 48 h. (**e**) Chilling injury index of *Cs*/*Cs* or *Cs*/*Cm* plants after chilling treatment for 0 h and 72 h. (**f**) *F*_v_/*F*_m_ in *Cs*/*Cs* or *Cs*/*Cm* plants after chilling treatment for 0 h and 48 h. Plants were pretreated with 50 μM ABA, 5 mM DMTU + 50 μM ABA, 0.1 mM DPI +50 μM ABA, or distilled water (control) for 12 h and then exposed to chilling stress. Data are presented as the means ± SD (*n* = 3). Different letters indicate a statistical significance between samples at *p* < 0.05.

**Figure 8 ijms-23-16057-f008:**
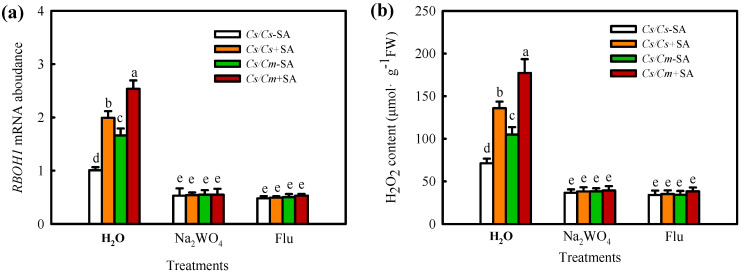
Effect of Na_2_WO_4_ and Flu on SA−induced *RBOH1* mRNA abundance (**a**) and H_2_O_2_ accumulation (**b**) in grafted cucumber plants. Plants were treated with 3 mM Na_2_WO_4_, 50 μM Flu, or deionized water, and 8 h later, 1.0 mM SA was sprayed. At 12 h after pretreatment, the plants were exposed to 5 °C for 6 h. Data are presented as the means ± SD (*n* = 3). Different letters indicate a statistical significance between samples at *p* < 0.05.

**Figure 9 ijms-23-16057-f009:**
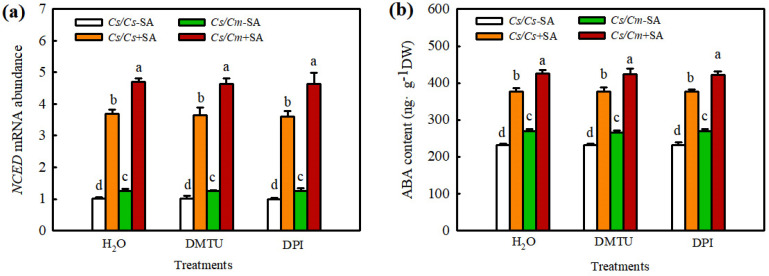
Effect of DMTU and DPI on SA−induced *NCED* mRNA abundance (**a**) and ABA content (**b**) in grafted cucumber plants. The plants were treated with 5.0 mM DMTU, 0.1mM DPI, or deionized water (H_2_O), and 8 h later, 1.0 mM SA was sprayed. At 12 h after pretreatment, the plants were exposed to 5 °C for 12 h. Data are presented as the means ± SD (*n* = 3). Different letters indicate a statistical significance between samples at *p* < 0.05.

**Figure 10 ijms-23-16057-f010:**
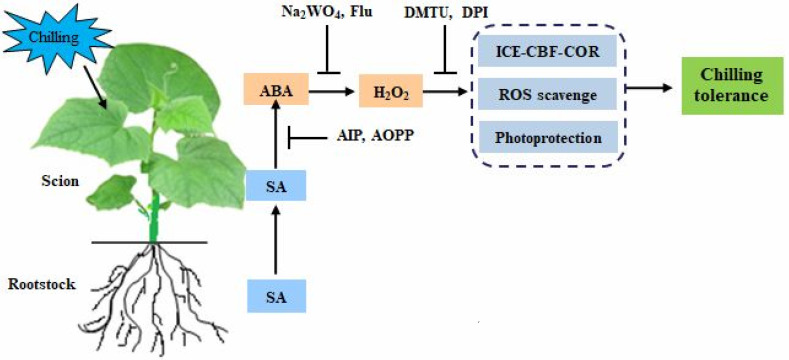
A model of SA-induced ABA and subsequent H_2_O_2_ accumulation in grafted cucumbers in response to chilling stress.

## Data Availability

The original contributions presented in the study are included in the article; further inquiries can be directed to the corresponding author/s.

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
