# Peer review of "Abscisic Acid Mediates Salicylic Acid Induced Chilling Tolerance of Grafted Cucumber by Activating H2O2 Biosynthesis and Accumulation"

_ijms, 2022, doi:10.3390/ijms232416057_

Round 1
Reviewer 1 Report
The work entitled “Abscisic acid mediates salicylic acid-induced chilling tolerance of grafted cucumber” fits with the aim of the journal “International Journal of Molecular Sciences”. The authors investigated the interaction of SA, ABA, and H2O2 in grafting-induced chilling response in cucumber. The manuscript is well-written and organized. However, the similarity rate is too high (43%). Consequently, I suggest to the authors to resubmit the manuscript after a significant reduction of the similarity rate.
Reviewer 2 Report
An interesting article. The manuscript addressed the interaction of SA, ABA, and H2O2 in grafting-induced chilling response in cucumber. It revealed the role of salicylic acid (SA) in grafting-induced chilling tolerance attributed to the ABA biosynthesis and the H2O2 accumulation in grafted cucumber leaves. The manuscript is well written. The results and discussion are informative. However, the manuscript cannot be accepted for publication in the current version because of the defects and concerns given below:
1. Title:
· I suggest the authors change the title to "Abscisic acid mediates salicylic acid-induced chilling tolerance of grafted cucumber by activating H2O2 biosynthesis and accumulation."
2. Abstract:
· I suggest the authors mention the results of the molecular experiments (gene expression) in the Abstract. The abstract implies that the authors did not perform any molecular experiments.
· Please mention the full names of all the abbreviated words, including Cm, SA, and Cs/Cm.
3. Introduction:
· Please replace "self-rooted" with "self-grafted" in the whole manuscript.
· Please double-check the reference writing style of MDPI in the whole manuscript.
· Page 2, Paragraph 4, Line 6, Please correct the spelling of leave as "leaves."
4. Results:
· Please replace "normal temperature" with "25 °C" in the whole manuscript.
· Page 2: Citing references in the results section is not preferred. I suggest that authors move Lines 1-2 and the last sentence (These data are consistent ....) in point 2.1. , page 2, to the discussion section.
· Page 3: Point 2.2: Citing references in the results section is not preferred. I suggest that authors move Lines 1-3 into point 2.2. , page 3, to the discussion section. Also, please, move the beginning of the sentence in line 3 and begin with the obtained results
· Page 9: Last paragraph, Line 2: Please mention the mRNA abundance for which gene as in figure 7a.
· Figures: In general, Please use the full names, not abbreviations, in the figure legends as much as possible.
§ Figure S1: Please move this figure to the supplementary materials section.
§ Figure S1:Please revise the spelling of temperature and sap in Figure S1.
§ Figure S1:There is no description of root and xylem sap sampling in the material and methods section. Please revise this point.
§ Figure 3: Gene names should be in italic. please revise in Figure 3d, and e.
§ Figure 3: Please try to show these two columns (Control, 25 C) in figure 3c by cutting the left Y axis into two segments. If these values are zero, How do the authors statistically analyze them?
§ Figure 4: Please, revise all the H2O2 in Figures 4 d and f. Please make it a subscript.
§ Figure 4:Please be consistent in writing Cs/Cs and Cs/Cm. Make them in Italic font in all the figures. For example, they look regular in Figures 4 e and f.
§ Figure 5: Please try to show these two columns (Control, 25 C) in figure 5 by cutting the left Y-axis into two segments. If these values are zero, How do the authors statistically analyze them?
§ Figure 5: There is no indication of a chilling injury index in the legend of figure 5, although it has three figures describing CI. Please revise.
§ Figure 6: Please add the statistical analysis (letters) in Figures 6a and b.
§ Figure 6 b, please consistently write the SA content unit in the whole manuscript.
§ Figure 6: Please try to show these two columns (Control, 25 C) in figure 6e by cutting the left Y-axis into two segments.
§ Figure 6: Please add the temperature treatments ( 5 C and 25 C) in Figures 6 e and f as in other figures in this panel.
§ Figure 6: Please be consistent in writing between the figures and the figure legend. For example, Electrolyte leakage and chilling injury index are written as the full name in the figures and abbreviation in the figure legend. I suggest the author use full names in the figure and legends.
§ Figure 7: Please add the statistical analysis (letters) in Figures 7a and b.
§ Figure 7: Is the statistical analysis in Figure 7e correct? For example, ABA (CS/Cm) took the "d" letter, and Control (25 C) took the same letter, "d," although the values are very far from each other.
§ Figure 7: Please try to show the two columns (Control, 25 C) in figure 7e by cutting the left Y-axis into two segments.
§ Figure 9: Please be consistent in writing the axes titles (bold or regular) in Figure 9 a and b.
§ Figure S2: Please be consistent in using colors in all the figures. For example, the authors used black and orange colors in all the figures, while in Figure S2, they used only black.
5. Discussion:
· Pages 12-13, last paragraph, lines (2-4): This sentence is not closely related to the manuscript subject. I suggest that authors delete or replace it with more relevant information. Is there any stress in that paper?
· Pages 13, paragraph 2, Line 5: Please mention the plant species, which is Tomato.
· Pages 13, paragraph 2, Line 6: Please replace "biotic stress" with "abiotic stress," as the subsequent sentences describe the abiotic stresses.
6. Materials and methods:
· Page 14, paragraph 1, line 4: Normally, Rootstock is germinated after the scion; as the rootstock grows faster than the scion, both can reach the suitable stage for grafting (cotyledon stage) at the same time. However, in the present manuscript, the authors grow the rootstock (Cm) first, then scion (Cs) was germinated 3 d later. It's not methodologically correct.
· Page 14, paragraph 1: What was the survival rate of the grafted plants?
· Point 4.2 Experimental design: Please mention how many replications and plants were used and the plant age (stage) in each experiment, as the authors conducted four separate experiments. Which Type of experimental design was used in the conducted experiments?
· Page 14, last paragraph: I suggest authors replace rooted with grafted in "self-rooted" and hetero-rooted." Please indicate the control treatment (self-grafted).
· Page 14, last paragraph, line 1: "plants with three leaves……." Do the authors mean "three true leaves"? Last line in the same paragraph (page 15): Please write the exact number. It is not preferred to say the plant number was 10-20 plants.
· Page 15, point 4.2.2., line 1: On which bases the authors chose these concentrations? Please add reference
· Page 15, point 4.2.2., line 3: What do the authors mean by 8/5 c? Please make it clear. "P" in phenotype should be a small letter
· Page 15, point 4.2.2., line 4: Please mention the exact measurements. It's not preferred to write "etc."
· Page 15, point 4.2.2., lines 8-9: Please mention that DMTU and DPI were used for H2O2.
· Page 15, point 4.3. Please indicate the tolerance grading criteria. And also, add ref. for the mentioned formula to measure the CI. This parameter is not clear to me.
· Page 16, point 4.8. Please be consistent in writing the gene names. They should be italic in the whole manuscript.
· Page 17: Figure 10: In the model, I suggest that authors indicate the SA, ABA, and H2O2 inhibitors and the H2O2 scavenger to draw the whole story.
· Page 19: Table S1, I suggest that authors remove the 5'-..........3' from the sequences and add it in the column title as "Primer sequences (5′→3′)". Also, the authors should show which primer is Forward (F) and which is reverse (R). It could be cleared by adding another column.
7. References: Please, carefully revise the references style. Scientific names should be in italic (such as Ref. No. 6 and 44)
Reviewer 3 Report
Dear authors,
Manuscript ijms-2046499 entiteled "Abscisic acid mediates salicylic acid-induced chilling tolerance of grafted cucumber" and authored by Yanyan Zhang , Xin Fu , Yiqing Feng , Xiaowei Zhang , Huangai Bi and Xizhen Ai targets a hot topic that is potentially very valuable for the journal readers. I appreciated reading this nice manuscript where experimets have been nicely designed and conducted. Conclusions are also supported by results. Infortunately few points need authors attention to meet the journal standards. I recommend few modifications before the manuscript could be accepted for publication:
1. Results : in Figure 1 the resolution of the figure have to be improved. In its current form it is hardly readable.
2. Conclusion : The conclusion is very weak I recommend to the authors to improve it mainly by highlighting their findings, drawing practical implications of their findings in the field and by pointing future research to go further in the field.
I will be delighted to read an improved version of this manuscript that I could recommend for publication
Best regards
Round 2
Reviewer 1 Report
Authors address all point of criticism, thus the manuscript can be published in present form.